# DFT Study of Adsorption Behavior of Nitro Species on Carbon-Doped Boron Nitride Nanoribbons for Toxic Gas Sensing

**DOI:** 10.3390/nano13081410

**Published:** 2023-04-19

**Authors:** Francisco Villanueva-Mejia, Santiago José Guevara-Martínez, Manuel Arroyo-Albiter, José Juan Alvarado-Flores, Adalberto Zamudio-Ojeda

**Affiliations:** 1Instituto Tecnológico de Pabellón de Arteaga, Carretera a la estación de Rincón de Romos, km 1, Pabellón de Arteaga 20670, Aguascalientes, Mexico; 2Department of Pharmacology, School of Exact Sciences and Engineering, University of Guadalajara, Boulevard Gral. Marcelino García Barragán 1421, Olímpica, Guadalajara 44840, Jalisco, Mexico; santiago.guevara@academicos.udg.mx; 3Instituto de Investigaciones Químico Biológicas, Universidad Michoacana de San Nicolás de Hidalgo, Morelia 58030, Michoacán, Mexico; manuel.arroyo@umich.mx; 4Facultad de Ingeniería en Tecnología de la Madera, Universidad Michoacana de San Nicolás de Hidalgo, Morelia 58040, Michoacán, Mexico; jjalvarado@umich.mx; 5Department of Physics, School of Exact Sciences and Engineering, University of Guadalajara, Boulevard Gral. Marcelino García Barragán 1421, Olímpica, Guadalajara 44840, Jalisco, Mexico; adalberto.zojeda@academicos.udg.mx

**Keywords:** reactivity, nanoribbons, adsorption, BNNRs

## Abstract

The modifications of the electronic properties on carbon-doped boron nitride nanoribbons (BNNRs) as a response to the adsorption of different nitro species were investigated in the framework of the density functional theory within the generalized gradient approximation. Calculations were performed using the SIESTA code. We found that the main response involved tuning the original magnetic behavior to a non-magnetic system when the molecule was chemisorbed on the carbon-doped BNNR. It was also revealed that some species could be dissociated through the adsorption process. Furthermore, the nitro species preferred to interact over nanosurfaces where dopants substituted the B sublattice of the carbon-doped BNNRs. Most importantly, the switch on the magnetic behavior offers the opportunity to apply these systems to fit novel technological applications.

## 1. Introduction

Toxic gases are major contributors to ambient air pollution and pose a serious threat to both human health and the environment. Rapid detection of these gases is crucial, and in some cases, they need to be neutralized to prevent harm, becoming harmless [1]. The use of low-dimensional gas-sensor systems for indoor and outdoor environmental monitoring is becoming increasingly prevalent [2]. Nitrogen-containing gases play a critical role in protecting the environment, promoting human health, and supporting biomedicine and agriculture [3]. Toxic gases, such as CO and NO, are particularly harmful and can cause severe health problems, including reduced oxygen delivery to vital organs, skin issues, and respiratory complications [1]. Exposure to even low concentrations of nitrogen dioxide in enclosed spaces can lead to respiratory complications and even death [4]. Additionally, ammonia can harm human health through tropospheric ozone formation, acidification of rain, and eutrophication [5]. Recent advancements in gas-sensor technology have allowed for the detection of NO_2_ and NH_3_ at the level of a single gas molecule on a thin-film material [6]. Theoretical analyses have demonstrated the effectiveness of graphene-based systems in detecting nitrogen-containing gases, and these promising results have prompted further experimentation and application of low-dimensional gas sensors [7,8]. Theoretical calculations have also shown that boron nitride systems, which are analog materials with similarities to graphene, can be utilized for gas sensing. Therefore, conducting theoretical calculations to determine the gas sensing properties of boron nitride systems could be a significant milestone in discovering a more effective sensing material.

The desire to downsize gas sensors while maintaining high sensitivity, selectivity, rapid response, and low cost has led scientists to investigate gas sensing systems based on novel nanomaterials [9,10,11]. Graphene, a pure two-dimensional crystal made of a *sp*^2^-hybridized carbon, has opened up a vast area of research exploring related low-dimensional geometries due to its exceptional properties, including high chemical stability, thermal stability, surface–volume ratio, carrier mobility, and fast response time [12,13,14,15,16]. Both experimental and theoretical investigations have been conducted on graphene’s applications in gas sensing and biosensing [17,18,19,20,21]. Such studies have sparked an increased interest in exploring related inorganic two-dimensional geometries for further research.

Hexagonal boron nitride sheets are another geometry that has recently gained significant research interest due to its analogical similarity to graphene [22,23,24]. Similar to graphene nanoribbons, boron nitride nanoribbons (BNNRs) can also be classified into zigzag and armchair configurations [25]. Previous studies have described the band gap behavior for each configuration of BNNR, and the introduction of functionalization and substitutional carbon doping has modulated their band gap, tuning their electronic properties from insulators to semiconducting materials [26,27]. The introduction of doped systems not only modifies the electronic properties of the boron nitride nanoribbon but also affects its intrinsic reactivity, describing the more interactive areas on the material and explaining the preferred interaction mechanism [28]. This makes BNNRs a promising material for novel technological applications, such as gas sensing. Theoretical DFT calculations have demonstrated efficient sensing capabilities of BNNRs for biosensing [29], alkaloid sensing [30], PH_3_ gas sensing [31], and NO reduction [32]. Therefore, addressing theoretical calculations may provide interesting results before developing an efficient gas sensor.

Some of the toxic gas molecules (NH_3_, NH_4_, CO, and NO_2_) show low sensitivity towards pristine GNR, which limits its potential use in the detection of individual gas molecules [33,34,35]. However, various reports discovered strong interactions between doped graphene and the same mentioned four gases, enhancing gas sensors [7,36,37,38]. Metal-doped graphene has also been shown to exhibit high sensitivity towards O_2_ adsorption [39], while defected graphene has been reported to have higher interactions with H_2_S than pristine graphene [40]. Furthermore, metal-decorated graphene has been found to have a high sensing performance towards NO, H_2_S, and HCN molecules, as the gas molecules bind firmly to the carbon structures [41,42].

In this study, we utilized first-principles methods based on density functional theory to investigate the electronic response of carbon-doped boron nitride nanoribbons upon exposure to toxic gas molecules, including NO_2_, NH_3_, and NH_4_. By analyzing the changes in their electronic properties, we evaluated the sensitivity of the carbon-doped BNNR sensor. Our findings suggest that these systems have great potential in the development of ultrahigh-sensitivity gas sensors.

## 2. Computation Details

The SIESTA code was used to perform all calculations for this work, with the projector augmented-wave method (PAW) describing the interaction between ionic cores and valence electrons [43]. The Perdew, Burke, and Ernzerhof parameterization (PBE) of the generalized gradient approximation (GGA) was used for the exchange-correlation functional [44]. The plane-wave basis-set cutoff energy was set to 450 eV, and the conjugate gradient method was used until the forces on each atom were lower than 0.01 eV/Å. The orthorhombic unit cell dimensions on the non-periodic axis were large enough to avoid interactions between images and edged hydrogen atoms, passivating the N and B atoms on the edges. The unit cell dimensions were optimized through the periodic dimension (along the *y*-axis (Appendix A)) until the cell total stress was close to a minimum, while a space of ~10 Å along the non-periodic directions was fixed. A Monkhorst-Pack grid of 1 × 20 × 1 k-points was used to sample the Brillouin zone in the reciprocal space, with a plane-wave cutoff energy of 450 eV. Spin-polarized calculations were performed for the carbon-doped BNNR nanostructures, and the total energy per unit cell was more negative in the spin-polarized approximation than in the non-polarized approximation (more than 2.27 eV of negative difference); these significant differences are in agreement with the literature. The results will be discussed in terms of the spin-polarized solution, with the total energy per unit cell being more negative than the non-polarized approximation. Therefore, nanoribbons with magnetic behavior will be analyzed (M-BNNR). In spin-restricted or -unrestricted calculations, closed shell systems are characterized by having doubly occupied molecular orbitals (restricted case), whereas in open shell systems, the spin degree of freedom makes it possible to obtain a complete set of orbitals for α-spin electrons and another set for β-spin electrons. The calculations were also used to describe systems that can be periodic in one dimension, with good agreement with experimental data.

## 3. Results

### 3.1. System Description

Firstly, the M-BNNRs considered in this work were built by passivating the edge N and B atoms with one hydrogen atom attached per edge atom. The selected size of these nanoribbons was 12 × 2, following a convention of size M × N, with M dimer lines along the width of the M-BNNR and N being the number of six-member rings along the periodic direction. The optimized geometries resulted in average bond lengths of 1.43, 1.45, and 1.08 Å for the pairs C-N, B-N, and C-H, respectively. Figure 1 provides the schematic representation of 12 × 2 M-BNNRs passivated by hydrogen atoms. In Figure 1a, the optimized structure of the nanoribbon shows a C atom substituting a N atom on the edge, whereas in Figure 1b, a C atom substitutes a N atom at the center of the nanoribbon. With a constant concentration of dopants equal to 1.78% and only one carbon atom on the BNNR, the relative position of the substitute atom shown in Figure 1 determines the identification for each possible arrangement, which is named P-M-BNNR, where P represents the relative position of the carbon atom on the nanoribbon (P = E or C), E indicating the edge and C the center of the M-BNNR.

Spin-polarized and non-spin-polarized solutions of the Kohn–Sham equations were analyzed, finding significant differences in the proposed systems. The doping effect in the system causes an unpaired number of electrons that, in contrast to a pristine BNNR, may be the reason for these significant differences. A previous study by Du et al. reported the spontaneous magnetization of BNNRs with a single C substitution [45].

### 3.2. System Electronic Properties

In this section, we discuss the band structure and total density of states of M-BNNRs, as presented in Figure 2. To obtain these figures, a Monkhorst-Pack grid with twice the density employed for energy calculations was used, and an integration of the Brillouin zone was performed. To improve the convergence of the integrals, a Gaussian smearing with a width of 0.01 eV for the levels was employed.

Figure 2b,c clearly show that the substitution of a N atom with a C atom results in an unoccupied band with a spin up contribution that shifts downwards from its original position near the Fermi level. In contrast, a pristine BNNR system exhibits insulator properties, with a band gap of approximately 4.5 eV, as shown in Figure 2a.

The magnetic behavior observed in the M-BNNRs can be attributed to the contribution of the C atom, which results in an unpaired number of electrons compared to the pristine system. Specifically, the substitution of a N atom with a C atom results in one less electron, which gives rise to a system with magnetic behavior.

### 3.3. Energetic Stability

In order to assess the energetic stability of the proposed systems, the cohesive energy (CE) was computed, and the results are reported in Table 1. The cohesive energy is defined as the amount of energy required to separate isolated atomic species of a solid, and a positive CE indicates a bound and stable state. Higher values of CE indicate more stable states. The computed CE values for a cubic BN and h-BN sheet are reported to be 7.12 and 7.09 eV atom^−1^, respectively, which are in good agreement with the size of the analyzed nanoribbon. It is theoretically expected that the cohesive energy decreases as the system size is reduced. The molar Gibbs free energy (*δG*) is also displayed in parenthesis in Table 1 and is given by
(1)$$\delta G=E\left(x\right)+\overset{n}{\underset{i=1}{\sum }}X_{i}\mu_{i}$$
where *E(x)* corresponds to the binding energy per atom and *X_i_* is the molar fraction of the proper atomic species (H, B, C, N), which satisfies ∑*X_i_* = 1. The chemical potential is approximated by the binding energy per atom of the ground state of H_2_ and N_2_ in the singlet state, the ground state of B_2_ in its triplet state, and the cohesive energy per atom of a graphene sheet. The energetic stability trend observed in the computed systems shows that doping at the edge is more energetically favorable than at the center of the ribbon, with the trend being pristine BNNR > E_M-BNNR > C_M-BNNR, indicating that the systems become increasingly energetically stable with dopant atoms on the edge rather that at the center of the nanoribbon.

### 3.4. Interaction of Toxic Gases on M-BNNRs

In this section, to further explore the potential of M-BNNRs in gas-sensing applications, the response to different molecules of interest was analyzed. While pristine graphene nanoribbons can detect these toxic molecules, doped graphene nanoribbons have shown improved gas-sensing capabilities due to the presence of dopant atoms. In the case of M-BNNRs, these toxic molecules can interact with the possible sites of the nanoribbons, leading to a corresponding change in the magnetic electronic behavior. This sensitivity to the adsorption of a single toxic molecule suggests that M-BNNRs could serve as highly responsive material for the detection of these harmful gases.

The molecular electrostatic potential analysis has provided additional insights into the interaction between the molecules of interest and the M-BNNRs. Specifically, the substitution of the C atom on the proposed site activates the surrounding N atoms, resulting in a more positive and larger site for interaction with the molecules. To study the interaction process, various initial configurations were proposed, taking into account the orientation and location of the molecules with respect to the activated area.

The interaction energies, optimized complexes, and corresponding modifications in the electronic properties were analyzed to understand the interaction process between the molecules and M-BNNRs. This information can be used to synthesize M-BNNRs that exhibit a high sensitivity toward the toxic molecules of interest and potentially develop them for use in gas-sensing applications.

Table 2 summarizes the interaction energies between the M-BNNRs and the analyzed toxic molecules. The interaction energy (*I_E_*) was calculated by using the following expression:(2)$$I_{E}=E_{complex}-\left(E_{P\_M-BNNR}+E_{M}\right)$$
where *E_complex_* is the computed energy of the interaction system, *E_P_M-BNNR_* represents the energy of the optimized magnetic nanoribbon, and *E_M_* designates the energy of the toxic molecule of interest, all of them in gas phase.

Table 2 suggests that the edge of the ribbon is more reactive and provides a more attractive site for the interaction of the studied molecules. The negative interaction energies indicate that the molecules are adsorbed on the surface of the M-BNNRs, and this adsorption process can modify the electronic properties of the system. The larger negative interaction energies observed for the E_M-BNNRs suggest that the substitution for the C atom at the center of the ribbon results in a less favorable site for interaction compared to the edge one. The modification of the electronic properties due to the adsorption of the studied molecules will be discussed in the next section.

## 4. Discussion

The final configurations of the interaction process between the systems are presented in Figure 3. Figure 3a,b demonstrate that the NO_2_ molecule can directly interact with the C dopant atom in the M-BNNR. These interaction complexes bind the N atom of the NO_2_ molecule to the C atom on the M-BNNR. The binding distances between the C-N atoms are reported to be 1.49 and 1.52 Å for the edge and center M-BNNR, respectively. In addition, a curve deformation on the M-BNNR for the C atom is observed, which may be due to the small size of the M-BNNR, where the cohesive energy is not sufficient to maintain the original six-member ring form.

Figure 3c,d illustrate the interaction of ammonium with the M-BNNR. In both systems, one H atom from the ammonium molecule binds to the C atom on the M-BNNR, resulting in an average bond distance of 1.12 Å between the C-H pair. Additionally, these calculations reveal the deprotonation for the remaining portion of the ammonium molecule, as evidenced by the reported distance between the bonded H atom and the nearest N atom. The separation distances between N-H pairs were found to be 2.34 and 2.21 Å for the edge and center M-BNNR, respectively. A curve deformation on the C atom can also be observed, although these deformations are less pronounced than the ones previously mentioned.

Figure 3e showcases the interaction between ammonia and E_M-BNNR, which represents the final and most favorable interaction observed in this study. The bonding between the species results in the most evident deformation. Initial configurations of the ammonia molecule oriented on top of the C atom were unresponsive, prompting a shift to the three nearest B atoms. The superior B atom exhibited the strongest attraction to the ammonia molecule, forming bonds described by B-N pairs with a separation of 1.62 Å. Similar analyses were performed on the C_M-BNNR, exploring initial configurations of the ammonia molecule on top of the C atom, the nearest B atoms, and even on the nearest N atoms. However, the interaction process was found to be unfavorable in all cases. In some instances, repulsion processes were observed at the nearest N atoms. Further studies are required to determine why the interaction process on this site is not favorable, as represented in Figure 3d.

Figure 4 shows the modifications in the magnetic electronic properties of the M-BNNRs following the interaction process. In all cases, with the exception of Figure 4f, the original magnetic behavior of the M-BNNR transitioned to a non-magnetic behavior after bonding with a molecule such as nitrogen dioxide or ammonia, a part of the original molecule, as in the case of ammonium. Following bonding onto the M-BNNR, the corresponding graphs describing the electronic properties now exhibit an electron pair in each band in both the band-structure graph and the local-density-of-states graph. Only Figure 4f retains an unpaired number of electrons in each band-structure and local-density-of-states graph.

The results of the present study demonstrated a stronger interaction between the M-BNNR and NO_2_ compared to a similar interaction reported by Bagheri and Peyghan, where they observed interaction separation distances of approximately 2 Å between a nanocone and the same molecule [46]. Additionally, a similar deformation behavior was observed in the study conducted by Esrafili in 2019, where the NO_2_ adsorption on a boron and nitrogen-doped graphene nanoribbon resulted in a slight curvature directly where the molecule interacted. The separation distance for their system was larger than 1.66 Å [37]. Moreover, the ammonium interaction observed in the present study involved a partial deprotonation of the molecule resulting in the separation of the molecule into ammonia and a hydrogen atom bonded to the M-BNNR. To our knowledge, this type of interaction has not yet been reported in the literature. Finally, the NH_3_ interaction on top of a zigzag graphene nanoribbon converged with a separation of 3.03 Å reported by E. Salih et al., more than double the separation reported in this study [47].

The effect of the carbon-dopant relative position in the same sublattice was also examined, while maintaining a dopant concentration of 1.78% of C atoms. Our findings are consistent with previous reports, which suggest that the edges of the M-BNNRs exhibit a higher interaction with nitrogenated molecules than the center part.

## 5. Conclusions

In this study, we conducted a first-principles investigation of the electronic properties of M-BNNRs using DFT. This study presented an investigation of the interaction between M-BNNRs and three nitrogenated molecules, namely nitrogen dioxide, ammonium, and ammonia.

Our study indicates that M-BNNRs can be utilized for the detection of nitrogenated molecules. By analyzing the shift from magnetic to non-magnetic behavior due to the interaction of these molecules with the M-BNNRs, we took the first step towards designing a low-dimensional gas sensor. The order of interaction in terms of the energy involved in the interaction of these nitro molecules is as follows: ammonium > ammonia > nitrogen dioxide.

To gain a better understanding of these processes, future research could investigate mixtures of these molecules, including their interactions with both the dopant atom and the delocalized π-systems.

## Figures and Tables

**Figure 1 nanomaterials-13-01410-f001:**
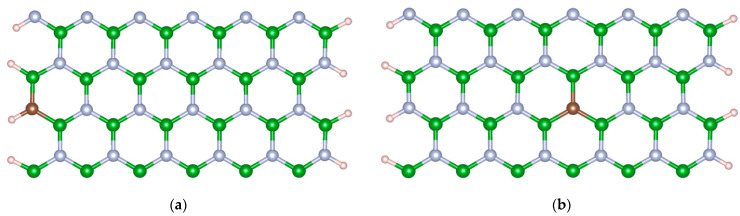
Schematic representations of (**a**) E_M-BNNR and (**b**) C_M-BNNR. The green sphere represents the B atom, grey sphere N atom, brown sphere C atom, and pink sphere H atom.

**Figure 2 nanomaterials-13-01410-f002:**
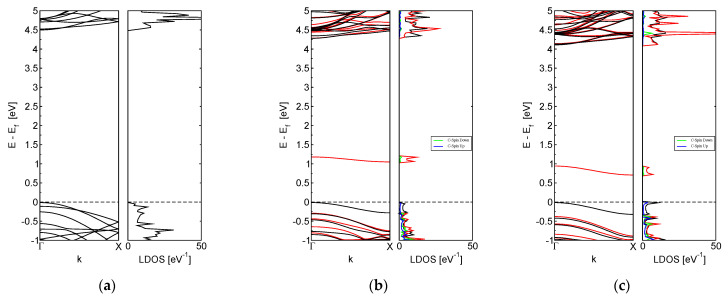
Electronic properties of the analyzed systems: (**a**) pristine BNNR, (**b**) E_M-BNNR, and (**c**) C_M-BNNR. Graphs represent the band structure with the corresponding total density of states.

**Figure 3 nanomaterials-13-01410-f003:**
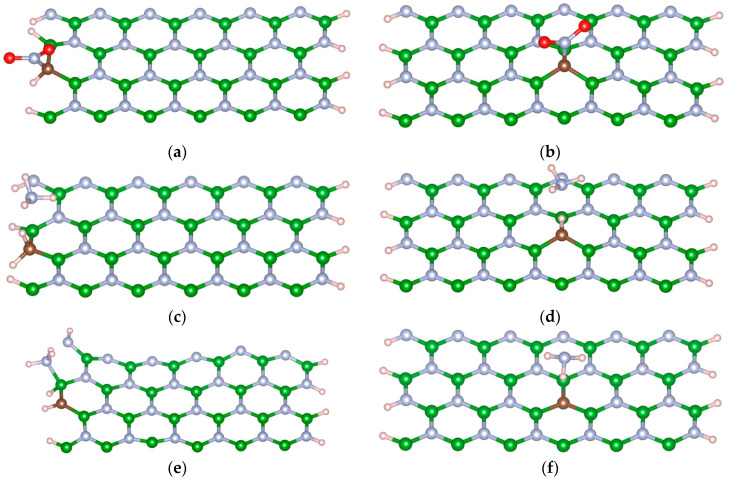
Schematic representation of the most stable interaction complex between M-BNNR and toxic molecules: (**a**) NO_2_ on the edge, (**b**) NO_2_ at the center, (**c**) NH_4_ on the edge, (**d**) NH_4_ at the center, (**e**) NH_3_ on the edge, and (**f**) NH_3_ at the center.

**Figure 4 nanomaterials-13-01410-f004:**
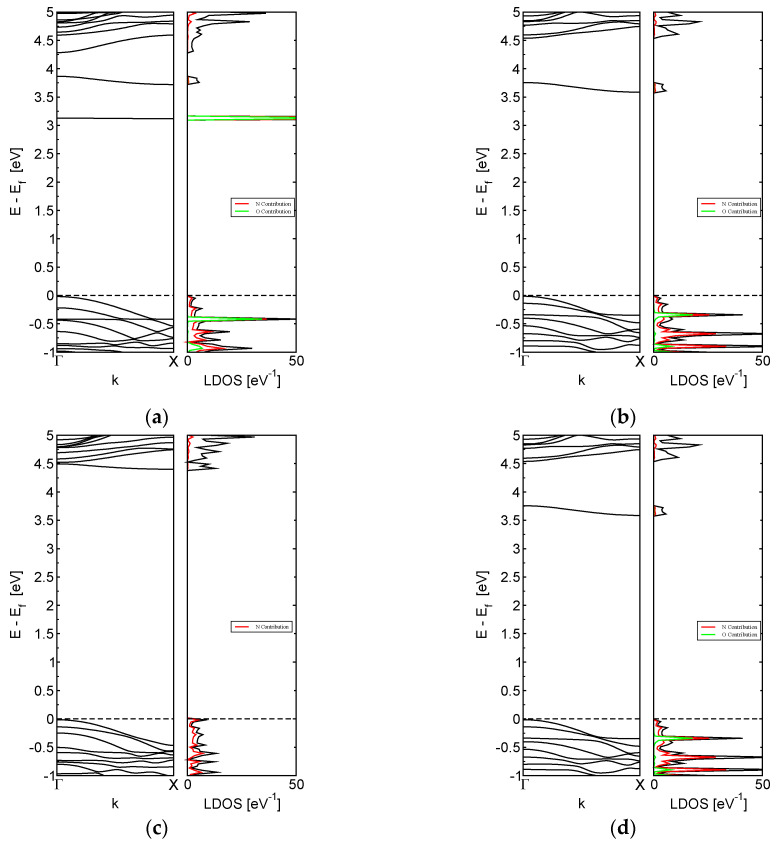
Electronic properties after the adsorption process of NO_2_ for (**a**) E_M-BNNR and (**b**) C_M-BNNR, NH_3_ for (**c**) E_M-BNNR and (**d**) C_M-BNNR, and NH_4_ for (**e**) E_M-BNNR and (**f**) C_M-BNNR. Graphs represent band structure with the corresponding local density of states.

**Table 1 nanomaterials-13-01410-t001:** Cohesive energy per atom (and Gibbs free energy) in eV of the pristine BNNR and M-BNNR.

M × N	Pristine	E_M-BBNR	C_M-BNNR
12 × 2	6.32 (−5.5)	6.30 (−5.39)	6.29 (−5.38)

**Table 2 nanomaterials-13-01410-t002:** Interaction energy of the most stable complex between the analyzed toxic molecules on the M-BNNR, measured in kcal/mol.

M-BNNR	NO_2_	NH_4_	NH_3_
E_M-BNNR	−50.04	−87.09	−80.47
C_M-BNNR	−44.23	−85.02	NR

## Data Availability

The data presented in this study are available on a reasonable request from the corresponding author.

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
