# Peer review of "DFT Study of Adsorption Behavior of Nitro Species on Carbon-Doped Boron Nitride Nanoribbons for Toxic Gas Sensing"

_nanomaterials, 2023, doi:10.3390/nano13081410_

Round 1
Reviewer 1 Report
Villanueva-Mejia et al., submitted the manuscript entitled “DFT study of adsorption behavior of nitro species on carbon doped Boron Nitride Nanoribbons for nano sensor applications” to “Nanomaterials (I. F = 5.719)”. This work requires major revision before acceptance.
1. Change the title with specificity. For example, instead of ‘Nano Sensor’ use ‘Toxic gas’ or ‘NO2, NH3 and NH4’. Current title is meaningless and led to loss of readability, because so many nano sensors available.
2. Introduction must be re-written pronouncing the motivation on “Why need DFT study with this material and why only NO2, NH3 and NH4 chosen? Also, state the importance of NO2, NH3 and NH4 and their toxic harm to the environment.
3. Result section is a mixture of discussion, author must shift those discussion points to the section 4. Discussion, which is not up to journal’s standard.
4. Conclusion and abstract section must be reframed as follows (A) in abstract state the importance of NO2, NH3 and NH4 and need of DFT investigation on sensitivity of Boron Nitride Nanoribbons towards those analytes and (B) in conclusion deliver the merits and certain limitations of this work along with future direction. In fact, currently both abstract and conclusion are without essential information.
5. Enhance the resolution of all figures. Especially, text font sizes of X, Y-axis and inner text of figures.
6. Update the reference section on the importance of NO2, NH3 and NH4.
Author Response
Please see the attachment, our corrected manuscript is under English Editing, it will be uploaded after MDPI English editing is completed.

Reviewer 2 Report
In this manuscript, the author considered carbon-doped boron nitride nanosheets for the sensing/separation of a few toxic molecules such as NH3, NO2, etc. They considered two different positions for doping (edge and center) and argued that the edged one is more efficient. Further, they have some discussion on the magnetic property of this system and how it changes with the adsorption. However, this reviewer thinks that the manuscript is very difficult to follow, and the computational methodology is not adequate and clear. The results discussed are not consistent and a thorough benchmark study is needed for further validation. The conclusion is not novel and not properly supported by the data in the results section.
I do not recommend this paper be published in Nanomaterials. My comments are given below to help the authors modify the manuscript.
- The authors need to improve their writing extensively throughout the whole manuscript. This reviewer finds it very difficult to follow the manuscript. Many sentences are very long with excessive use of commas (,) which makes them very complicated to understand. I recommend revising the language in the manuscript and using small sentences to make it simpler.
- I recommend the authors provide coordinates of the unit cell (with lattice vector for the periodic box) for both E_M-BNNR. And C_M-BNNR. It would be curious to know if this is one-dimensional or two-dimensional in the periodic calculation. In Figure 1 the side (horizontal) terminal B, N, and C are passivated by H but not the up-bottom (vertical) B and N atoms. I guess the vertical direction is the expanding direction. If so, why the author reduces this two-dimensional(2D) material to a one-dimensional (1D) system? Please clarify the modeling of the nanosheet.
- The choice of high k-point 1*20*1 is not very clear to me. Does the author have and benchmark study or a reference behind choosing this grid? Also, the authors mentioned the use of both 450ev and 500 eV as plane-wave cutoff values. Which one is used for the results described in the manuscript?
- The spin-polarized calculation is mentioned to give lower energy than the non-polarized one. It is important to know how much energy difference the author got from the two methods on the C_M-BNNR.
- Page 4, line 53; What is the energy difference between E_M-BNNR and C_M-BNNR? That value needs to be mentioned to get an idea about the relative stability.
- The results in Table 2 do not seem to be logical to this reviewer. First of all, the binding energy of +749.77 kcal/mol for NH3 on C_M-BNNR is impossible, there must be some error in the calculation. Also, I don’t get how the binding energy of NH4+ (has to be a “+” sign as the formal charge) is so high even higher than NH3. NH4+ does not have any unpaired electrons to bind and N is four-coordinated so reluctant to form any bonding interactions.
- Figure 3(c,d), The NH4+ is no longer adsorbed. It is dissociated to form NH3, and the Caron is protonated. Hence, the energy reported by the author is not interaction energy rather it is reaction energy. The nanoribbon is protonated. I am curious why the author chose NH4+ as a molecule as it is positively charged and lack of coordinating electrons.
- The conclusion that the magnetic property is lost when the toxic molecule is adsorbed is expected. It would be nice if the authors calculate the spin density which should be mostly localized around the carbon in the bare nanosheet. In the adsorbed structure the magnetism is lost as the carbon is bonded with the molecules.
- The author must provide all the optimized coordinates along with the lattice vectors in a supplementary document.

Author Response

(The authors gave the same response as above.)

Round 2
Reviewer 1 Report
Author addressed all the queries. But, in the title replace "Toxic gas applications" to "Toxic gas detection" or "Toxic gas sensing"
Author Response
Thank you for the comments and suggestions. We have changed the title of our manuscript to “DFT study of adsorption behavior of nitro species on carbon doped Boron Nitride Nanoribbons for Toxic gas sensing”.

Reviewer 2 Report
The author worked on the language extensively and made the manuscript suitable for publication now. Also, they addressed my comments. At this stage, the manuscript can be published in Nanomaterials.
Author Response
Thank your for the comments that made us improve our manuscript, please see the final version.